# Eminuscent phase in frustrated magnets: a challenge to quantum spin liquids

S. V. Syzranov [1✉] & A. P. Ramirez [1]

A geometrically frustrated (GF) magnet consists of localised magnetic moments, spins, whose orientation cannot be arranged to simultaneously minimise their interaction energies. Such materials may host novel fascinating phases of matter, such as fluid-like states called quantum spin-liquids. GF magnets have, like all solid-state systems, randomly located impurities whose magnetic moments may "freeze" at low temperatures, making the system enter a spin-glass state. We analyse the available data for spin-glass transitions in GF materials and find a surprising trend: the glass-transition temperature grows with decreasing impurity concentration and reaches a finite value in the impurity-free limit at a previously unidentified, "hidden", energy scale. We propose a scenario in which the interplay of inter-actions and entropy leads to a crossover in the permeability of the medium that assists glass freezing at low temperatures. This low-temperature, "eminuscent", phase may obscure or even destroy the widely-sought spin-liquid states in rather clean systems.

[1] Physics Department, University of California Santa Cruz, Santa Cruz, CA 95064, USA. ✉email: syzranov@ucsc.edu

In magnetic systems, frustration[1,2] is essential to avoid long-range order and realise theoretically predicted quantum-spin-liquid states, of potential use for quantum computation. Frustration may come from tuned interactions, as in potential Kitaev materials[3], or from the geometry of the lattice, as in a broad class of materials called geometrically frustrated (GF) materials, the focus of this work. As a result, spin liquids are actively sought in materials with antiferromagnetic (AF) interactions on geometrically frustrating lattices[4], exemplified by the pyrochlore, Kagome and triangular structures.

Whereas spin liquid theories assume perfect lattices, real materials possess quenched disorder, such as vacancies or impurities, in varying degrees. Such disorder may fundamentally alter the nature of the ground state in geometrically frustrated systems by inducing a disordered state, possibly precluding the formation of a spin liquid.

Here we first address the disorder-induced spin-glass-freezing in GF materials and compare it with conventional spin-glass (SG) transitions in non-GF materials. The nature of disorder and the medium between impurities or vacancies are quite different in GF and non-GF SGs. The medium between impurities in non-GF systems is non-magnetic, while the medium in GF systems consists of atomic spins. This usually leads, for example, to a large quadratic in temperature specific heat $C(T)$ in GF SGs[5,6], in contrast with the impurity-dominated linear-in-temperature $C(T)$ found in non-GF SGs[7]. Defects in GF systems are normally represented by magnetic atom vacancies, while impurities in non-GF SGs are magnetic atoms.

Despite these differences, spin glasses in GF and non-GF systems share a number of common properties. Both classes of systems are believed to display spin-glass transitions in the same universality class[8] and both: (i) display a cusp in magnetic susceptibility $\chi(T)$ at the glass-transition temperature $T_g$; (ii) display hysteresis in $\chi(T)$ below $T_g$[5,9]; (iii) show a divergence in the non-linear susceptibility at $T_g$[5,10]; (iv) show no specific heat anomaly at $T_g$. Also, increasing the amount of quenched disorder (defects) in both GF and non-GF systems is commonly believed, and confirmed by numerous models, to favour spin freezing and increase the glass-transition temperature, regardless of the nature of disorder, (see, for example, refs. [8,11–13]).

In this paper, we are questioning conventional understanding of the response of GF systems to defects. We analyse the available experimental data for $\chi(T)$ and $T_g$ as a function of the density of randomly located defects and demonstrate that the phenomenology of the glass transition in GF and non-GF systems is dramatically different. In both classes of materials, $\chi(T)$ increases with increasing the density of defects, magnetic impurities in non-GF systems and vacancies in GF systems. In GF systems, however, this trend is accompanied by a decrease of the critical temperature $T_g$, in contrast with non-GF SGs as well as with common intuition. Furthermore, extrapolating $T_g$ in GF materials to the limit of zero vacancies gives a finite value, $T^*$, far below the Weiss constant $\theta_W$. Since vacancies represent the main source of disorder in GF materials, this suggests a transition at $T^*$ in pure GF systems.

This observation calls into question the observability of quantum spin liquids in GF materials since all solids possess some disorder and an SG state may be anathema to a spin liquid. In this paper, we propose a scenario of the transition, consistent with the available experimental data, in which the interplay of interactions and the entropy of the impurity spins leads to a crossover in the effective magnetic permeability of the medium, which in turn drives the transition.

## Results

We analyse and compare experimental data on the glass transition for GF and non-GF systems with varying degrees of

quenched disorder. For the former, we consider only strongly frustrated GF systems, i.e. $f = \theta_W/T_g 10$[14].

Unlike in non-GF systems, where disorder scales with atomic spin density, in GF materials disorder can arise from different sources, the most common of which are spin vacancies and structural distortions. For most of the materials considered here, disorder was varied by substituting a non-magnetic atom for a magnetic atom. Such a spin vacancy is generally considered to be the degree of freedom that freezes at $T_g$–the vacancy induces a shielding effect among its neighbouring spins, thus creating a composite spin-like variable, a "quasispin" or a "ghost spin" (see, e.g.[15,16]). In a few systems, the structure itself creates disorder, an extreme example of which is $Y_2Mo_2O_7$, where a dynamic Jahn-Teller transition, driven by the $Mo^{4+}$ ion, induces a (disordered) orbital dimer phase at its $T_g$, thus leading to bond disorder among the spins[17]. When this native disorder is increased by substitution of non-magnetic $Ti^{4+}$[18], the behaviour, to be discussed below, is similar to spin vacancy introduction in other GF systems. Another perhaps less extreme example is $Na_4Ir_3O_8$, where incomplete occupancy of $Na^{1+}$ on the $A$-site[19] likely leads to a random strain field that may induce randomness in the $Ir^{4+}$ spin exchange interactions. The other systems considered here do not possess such obvious defects but may, in synthesis, acquire disorder that exists below typical X-ray diffraction detection limits.

To reveal universal trends in GF and non-GF systems, which are independent of the nature of disorder, we plot in Fig. 1 $\chi(T)$ as a function of $T$ for various amounts of disorder (upper panel) and $\chi(T_g)$ as a function of $T_g$ (lower panels). The upper panels demonstrate a dramatic difference between representative non-GF and GF systems – while $\chi(T_g)$ increases with $T_g$ for the non-GF example, $\chi(T_g)$ decreases with $T_g$ for the GF example. In the lower panels of Fig. 1 we show the dimensionless $\chi(T_g)$ vs. $T_g$, using all available experimental data for families of compounds with variable disorder[5,9,18–30] (details on methodology are provided in the SI). We emphasise that, since different reviewed compounds have different microscopic sources of quenched disorder (to which impurity atoms contribute but of which they may not be the only source), the observed trends in the behaviour of $\chi(T_g)$ vs. $T_g$ are universal.

The second representation of the data (Fig. 2a) considers, for GF systems, the behaviour of $T_g$ not in relation to $\chi(T_g)$ but as a function of a known amount of disorder, specifically the nominal spin vacancy concentration, $x$. Although spin vacancies may not be the only source of disorder, as in $Y_2Mo_2O_7$, decreasing $x$ results in decreasing the total disorder strength, at least for sufficiently small $x$. In this linear $x$ vs. log $T_g$ plot, we see that the glass-transition temperature increases with decreasing disorder, strongly implying that $T_g$ reaches a finite value, $T^*$, in the pure system. We note that since no other substantial sources of disorder distinct from spin vacancies are known in the reviewed materials, except for $Y_2Mo_2O_7$ as discussed, the extrapolation to $x \rightarrow 0$ gives the perceived transition temperature, $T^*$, in pure materials. Since $T^*$ is an apparent property of the pure compound, and since it is at least an order of magnitude smaller than $\theta_W$, and has not been described via a microscopic theory, we refer to this as a "hidden" energy scale. Values for $T^*$, $\theta_W$, and other materials characteristics are compiled in Table 1.

While susceptibility yields information about the $q = 0$ response, other measurements, such as specific heat, muon relaxation, and neutron scattering, shed light on the spin configuration in the vicinity of $T^*$. Neutron scattering is especially informative since it shows, for the systems that approach $x = 0$, i.e. those that most closely approach "pure", a rapid rise in the intensity of short-range elastic scattering on approaching $T^*$ from above, and little temperature dependence below $T^*$[31–33]. Thus,

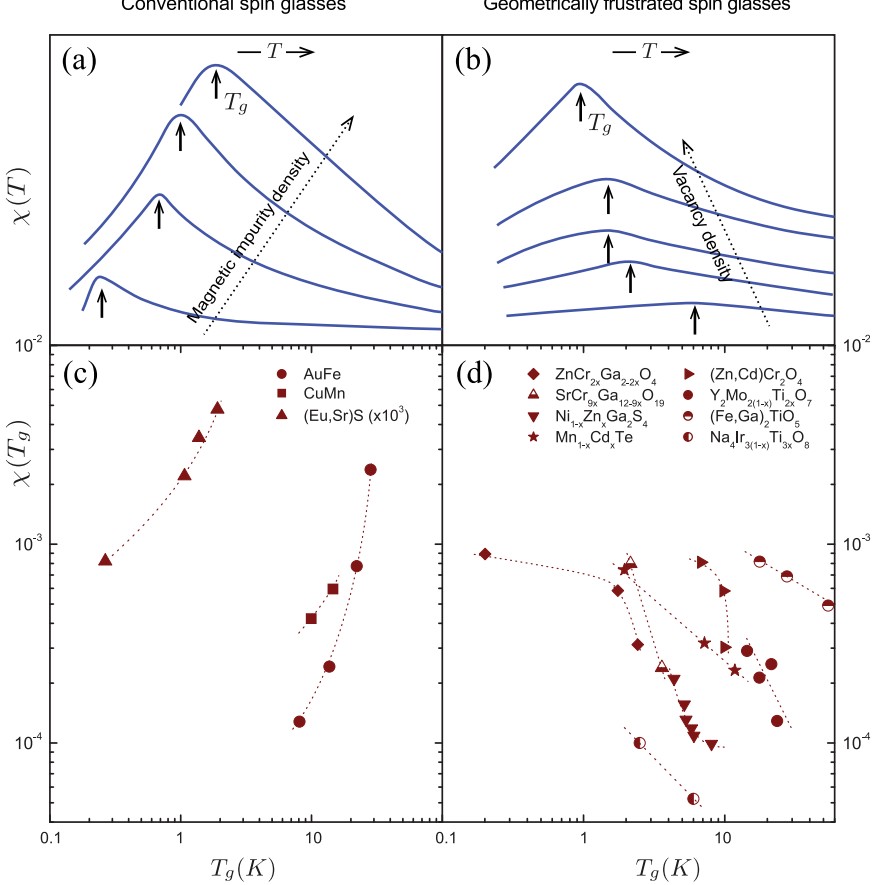

**Fig. 1 Behaviour of susceptibilities, $\chi(T)$ and $\chi(T_g)$ in conventional and geometrically frustrated systems. a, b** The susceptibilities $\chi(T)$ of $(Eu,Sr)S$[20] and $NiGa_2S_4$[21] as a function of temperature for various vacancy concentrations. The lines have been rendered from digitized data in the referenced publications and the $T_g$ values are plotted in the lower panels. **c, d** The susceptibilities $\chi(T_g)$ at $T_g$ vs. $T_g$ for conventional[22,23] and frustrated spin glasses. The reference sources for the GF data are provided in Table 1.

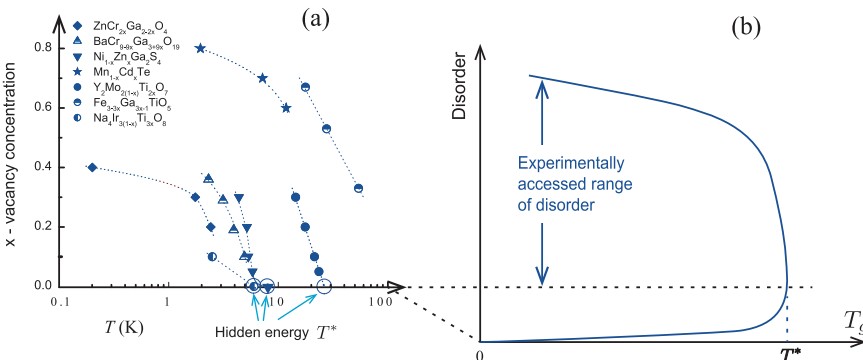

**Fig. 2 The dependencies of the glass transition temperatures $T_g$ on the amount of disorder in geometrically frustrated systems. a** The dependencies for real materials following from the available experimental data analysed in this work. The non-vanishing of $T_g$ in the limit of vanishing disorder for $NiGa_2S_4$, $Na_4Ir_3O_8$ and $Y_2Mo_2O_7$ suggests the existence of a hidden energy scale $T^*$. **b** The dependence proposed here for a broader range of disorder. The upper part of the curve ("experimentally accessed range of disorder") illustrates the experimental data analysed in this work. The lower part, for weaker disorder, is conjectured.

$T^*$ coincides with the development of short-range order among atomic spins on cooling. This short-range order, also revealed as a broad peak in specific heat, is clearly a property of the pure system. Moreover, unlike systems of reduced dimensionality where the temperature at which short-range order occurs reflects the magnitude of the dominant exchange interaction, no such microscopic origin for $T^*$ is presently known for GF systems. Absent a microscopic model, however, it is possible to create a framework phenomenology of this short-range ordered state, as described later.

In order to fully appreciate the implications of the GF/non-GF dichotomy shown in Fig. 1, one needs to consider the degrees of freedom that govern the glass freezing and the magnetic susceptibility in these classes of materials. For the non-GF systems there can be no doubt that the degrees of freedom are atomic spins, e.g. Mn atoms in a Cu host, as alluded to above[11,12,20,22,23].

**Table 1 Strongly geometrically frustrated magnets and their magnetic lattice, atomic spin (S), AF Weiss constant $\theta_W$, glass transition temperature $T_g$, frustration parameter $f = \theta_W / T_g$ and the hidden energy scale $T^\star$, for compounds pure enough to allow an estimate.**

| GF compound | lattice | S | $\theta_W$ (K) | $T_g$ (K) | f | $T^\star$ | Ref. |
|---|---|---|---|---|---|---|---|
| $ZnCr_{1.6}Ga_{0.4}O_4$ | spinel | 3/2 | 115 | 2.4 | 48 | n.e. | 16,24 |
| $Zn_{0.95}Cd_{0.05}Cr_2O_4$ | spinel | 3/2 | 500 ± 20 | 10 | 50 | n.e. | 27,47 |
| $SrCr_8Ga_4O_{19}$ | Kagome (layered) | 3/2 | 515 | 3.5 | 147 | n.e. | 5,25 |
| $BaCr_{8.1}Ga_{3.9}O_{19}$ | Kagome (layered) | 3/2 | 695 | 4.9 | 142 | n.e. | 48 |
| $NiGa_2S_4$ | triangular (layered) | 1 | 80 | 8.0 | 10 | 8 ± 0.2 | 6,21 |
| $Mn_{0.53}Cd_{0.47}Te$ | fcc | 5/2 | 292 | 15 ± 3 | 19 | n.e. | 26 |
| $Na_4Ir_{3(1-x)}Ti_{3x}O_8$ | hyperkagome | 1/2 | 570 | 6.0 | 100 | 6.0 | 19,29 |
| $Y_2Mo_2O_7$ | pyrochlore | 1 | 200 | 23.5 | 8.5 | 25 ± 1 | 9,18 |
| $(Fe,Ga)_2TiO_5$ | pseudobrookite | 5/2 | 900 ± 10 | 55 | 16 | n.e. | 28 |

Compounds for which $T^\star$ is not estimated due to insufficient disorder range are indicated by n.e. Entries not having an error estimate can be considered accurate to one digit in the least significant place. The compounds listed are the lowest-disorder members of their respective dilution series, data for which can be found in the references.

For GF systems, however, there are two options for magnetic degrees of freedom: (1) "quasispins"[15,16] forming around vacancies and acting as free magnetic moments or (2) the bulk spins away from the vacancies. The growth of the susceptibility $\chi(T)$ in GF systems with increasing the density of vacancies, similar to the growth of $\chi(T)$ with increasing the density of magnetic moments in non-GF systems, is evidence in favour of the quasispin scenario, which, however, does not explain the behaviour of $T_g$. The decrease of $T_g$ with increasing the vacancy concentration, on the contrary, suggests an essential role of the bulk degrees of freedom: as vacancies dilute the bulk spins, they lower the glass-transition temperature, provided the transition is driven by the bulk spins. At the same time, vacancies would decrease the contribution of the bulk spins to $\chi(T)$, which is observed to grow with vacancy density. Therefore, any explanation, including microscopic theories, of the behaviour of GF SGs must involve both the quasispin and bulk degrees of freedom and their interplay and be constrained by the equation $d\chi(T_g)/dT_g < 0$.

## Discussion

As discussed above, the observed increase of $T_g$ with decreasing disorder, to which vacancies contribute but of which they may not be the only source, is in sharp contrast with numerous models of spin glasses, including GF glasses[8,11,12] (we confirm, by a microscopic calculation, in[13] that $T_g$ vanishes in the limit of vanishing disorder strength for quenched disorder and scales as $T_g \propto n_{imp}^{1/2}$ if determined by sparsely located vacancies with the average density $n_{imp}$). Furthermore, this trend implies the existence of a glass transition in a disorder-free system, at a temperature given by the hidden energy scale $T^\star$.

A naïve explanation for the hidden energy scale in GF systems is the presence of non-vacancy sources of quenched disorder, such as random strain in $Y_2Mo_2O_7$, that could drive the glass transition in the zero-vacancy limit. However, the hidden energy scale has the same order of magnitude, $T^\star \sim 10K$, in all GF compounds (with the exception of $Fe_{3-3x}Ga_{3x-1}TiO_5$, in which $T^\star \sim 100K$ and the transition is apparently dominated by clusters of ~30 $Fe^{3+}$ spins rather than by single-vacancy spins[30]) Since the other compounds do not have significant known sources of disorder distinct from vacancies, this magnitude of the hidden energy scale requires an explanation.

Another possibility is a glass-freezing transition in the absence of quenched disorder[34]. Such a possibility has never been demonstrated rigorously but would be consistent with the saturation of the glass-transition temperature to a finite value $T^\star$ in the disorder free limit.

It is also possible that the GF medium undergoes a sharp crossover at a scale $T^\star$, determined only by the properties of the disorder-free GF medium, which drives the glass transition in the presence of a small residual number of vacancies (possibly undetected in experiments) or non-vacancy disorder. Below, we consider in more detail the latter scenario, which we believe to be the most likely, and which is consistent with the neutron-scattering data[31–33]. We assume that the transition temperature will eventually vanish in the limit of zero disorder and that GF systems studied to date have residual disorder (in the form of defects or other sources). A sharp crossover in the medium, as a function of temperature, can trigger a transition to the glass state with the temperature $T^\star$ determined by the properties of the medium and weakly dependent on the strength of the residual disorder for experimentally available systems.

The above discussion raises the question of what the transition at $T^\star$ in the pure system represents. At $T^\star$, the system transitions or crosses over from a high-temperature state with suppressed correlations between impurity atoms to a low-temperature phase in which the long-range entropic interactions are essential and assist the freezing of the impurity degrees of freedom. We refer to the latter phase as an "eminuscent", derived from the Latin "ex" (away from) and "manus" (hand), with "eminusque" being Latin for "long range".

The existence of the hidden energy scale suggests that a spin glass phase may be inescapable in real materials, even if they can be made significantly purer, which raises the important question of whether a quantum spin liquid can coexist with a spin glass – a question that is beyond the scope of the present work – and if not, whether spin liquids are achievable in GF systems.

The eminuscent-phase scenario seems, at first glance, to contradict experiments in some s = ½ GF systems[4,35,36], such as such as Herbertsmithite[37,38], $H_3LiIr_2O_6$[39] and $Cu_3Zn(OH)_6FBr$[40] that show no signs of spin freezing at temperatures down to 50 mK, and possess features not inconsistent with a spin liquid interpretation. This may be attributable to their effectively 2D nature, which precludes a glass transition at a finite temperature (see Supplementary Information): a system may already be in a glass state for the entire range of accessed temperatures or disallow for the glass phase (e.g. 2D Ising models with bond disorder[41]) .

To sum up, in this paper, we have analysed known SG transitions in GF magnets and show that they differ qualitatively from those in conventional glasses. The freezing temperature of such SG transitions grows with decreasing the concentration of vacancies, the main source of disorder in GF magnets, and reaches a finite value, the "hidden energy scale" in the perceived

disorder-free limit. This observation calls into question the achievability of quantum spin liquids in GF magnets. Indeed, if a glass transition occurs at a temperature determined by the properties of the medium and not by the amount of disorder, a system may always be entering the spin-glass state instead of becoming a spin liquid. We emphasise that such a possibility would be peculiar to materials with GF lattices and does not concern systems with bipartite lattices, such as the Shastry-Sutherland[42] and Kitaev's honeycomb[3] models, in which frustration is achieved via tuning of the interactions and not via the lattice structure.

The results in this work call for a thorough theoretical investigation of the role of quenched disorder in GF systems. While weak disorder is perturbatively irrelevant in a GF material[43], non-perturbative fluctuations of the locations of the impurities and inter-spin couplings may still result in a glass state even for very small amounts of disorder.

We have also proposed a phenomenological scenario for the hidden energy scale. We conjecture that the SG transition temperature vanishes at strictly zero disorder. However, for available impurity concentrations to date, glass freezing is driven by the crossover in the magnetic permeability rather than by the amount disorder. As a result, an extrapolation of the SG transition temperature in existing materials to the limit of zero disorder (vacancy density) leads to a finite value. The phenomenological picture of the SG transition developed here calls for a thorough investigation of the critical features of the transition and for synthesis of cleaner GF materials. Another question which deserves a thorough investigation is the possibility of a spin-freezing transition in a GF material in the absence of quenched disorder. We leave the investigation of such a possibility to future studies.

## Methods

To illustrate how the discussed crossover in the GF medium may occur, we consider a model of a spin ice or a system with vector spins in the "Coulomb phase"[4,44–46]. In such a description, the medium is mapped to classical Coulomb fields $\mathbf{B}$ and a system of charges (monopoles), topological defects that may be activated thermally or created by impurities. The spin medium between the defects determines the interaction energy $E_{int} = \frac{1}{\mu(T)} \sum_{i,j<i} \frac{Q_i Q_j}{|r_i - r_j|}$ between monopole charges $Q_i$ via the effective permeability $\mu(T)$.

By introducing the characteristic energy $\widetilde{T}$ of the static field $\mathbf{B}$ per spin (arising, e.g. from the dipole-dipole interactions or further-neighbour interactions), we arrive at the permeability (see[13] for details)

$$\mu(T) = \frac{\widetilde{T}}{\widetilde{T} + T} \qquad (1)$$

that exhibits a crossover from a constant value $\mu = 1$ at $T \ll \widetilde{T}$ to a strongly suppressed value at $T \gg \widetilde{T}$. The crossover in the permeability, $\mu(T)$, leads to a crossover in the ratio $\delta E/T \propto \left[\mu(T) T\right]^{-1} = \frac{T + \widetilde{T}}{\widetilde{T} T}$ of the fluctuation of the interaction energy of impurity spins to the temperature. In weakly disordered samples, this ratio is small at high temperatures $T \gg \widetilde{T}$ and grows rapidly for $T \ll \widetilde{T}$, thus favouring the glass state. In the presence of quenched disorder, the crossover in the magnetic permeability (1) will drive a glass transition at a temperature $T^*$ on the order of $\widetilde{T}$. While we considered a model of a spin glass here, we expect that such crossovers exist generically in GF media, with the details of the function $\mu(T)$ depending on the microscopic details of the Hamiltonian.

We suggest, therefore, that the SG transitions in GF materials observed to date are driven by the GF medium, with the transition temperature $T^* = \widetilde{T}$ weakly dependent on the amount of disorder for the lowest achieved concentrations of impurities. The dependence of the critical temperature on disorder implies a region of higher purity, depicted in Fig. 2b, that has not yet been probed. A careful verification of the scenario of the glass transition proposed here will require a thorough experimental investigation of the details of the transition and advances in the synthesis of clean GF systems.

## Data availability

The Data used to generate Figs. 1, 2 and Table 1 are available from the cited articles published by the Physical Society of Japan, American Physical Society and Institute of Physics.

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

## Acknowledgements

We have benefited from useful discussion with C. Batista, S. Brown, J.T. Chalker, S. Kivelson, S.-H. Lee, R. Moessner, S. Profumo, A.W. Sandvik and A.P. Young. APR would like to acknowledge support by the U.S. Department of Energy Office of Basic Energy Science, Division of Condensed Matter Physics grant DE-SC0017862.

## Author contributions

Both authors contributed significantly to the analysis of the data, calculations presented in this work and writing the manuscript.

## Competing interests

The authors declare no competing interests.
