## [Peer Review File · Nature Communications]

Reviewers' Comments:

Reviewer #1:

Remarks to the Author:

The authors have given detailed responses to the various points raised in the previous round of review. Many points are matters of differences of opinion that are unlikely to be resolved during further rounds of review. My sense from my own reading, the responses provided, and from the reviews from and responses to the other referees is that many of the ideas presented here will be considered interesting yet contentious within the community. I am happy to suggest that the paper be published.

Reviewer #2:

Remarks to the Author:

In this manuscript, Syznarov and Ramirez, compare the properties of geometrically frustrated spin glasses with conventional spin glasses using a collection of susceptibility data assembled from the literature. I have reviewed a previous version of this manuscript when it was under consideration at Nature Physics and have now read the revised version as well as the authors responses. The authors show that in geometrically frustrated spin glasses, a general trend can be observed where as the magnetic sublattice is diluted the freezing transition is suppressed in temperature, while the susceptibility magnitude increases.

In my previous report, I commented on the use of the term defects and disorder to describe what I feel is most accurately labelled as magnetic dilution. The manuscript still paints the overwhelming picture that the leading effect of replacing a magnetic ion with a non-magnetic ion is disorder. See for example the paragraph that starts at line 48. The effect of magnetic dilution is completely disregarded and never addressed and is extremely important to interpreting the trends discussed here. Further, the use of the term defects is still misleading, particularly in the context of Fig 1, where it appears in panel (a) and (b) with opposite meaning. Putting aside semantic differences, my primary criticism does not relate to the choice of terms but rather the fact that the effect of dilution (or whatever term the authors prefer) – meaning the reduced connectivity of the magnetic lattice – is never addressed.

The other main finding of this study is that, in the systems surveyed, the magnitude of the anomaly at T_g is increasing as a function of dilution/disorder. I agree that this is an interesting observation but again there are many conflating factors. One key factor is that dilution by reducing connectivity, alleviates geometric frustration. In the least diluted samples (the lowest defects in the authors terms), the behavior is closest to the ideal spin liquid limit where a larger fraction of the spins remain dynamic (as has been seen in other probes such as the persistent spin dynamics seen in μ SR). Then, as magnetic ions are replaced, frustrated is alleviated and the magnet becomes more conventional, resulting in more spins participating in the freezing transition and therefore enhancing the strength of the susceptibility anomaly. Again, I do think this is an interesting observation but because dilution and reduction of frustration are occurring simultaneously, I remain unconvinced that any single or universal effect related to disorder can be isolated.

In the final section of the paper, titled "Online methods" the authors present a picture for the behavior described above. The scenario being presented is vague and insufficiently detailed. The existence of frustrated materials without a spin glass transition is handwaved and relegated to the supplemental material.

Even if all the above criticisms were addressed, I would still find the format of this paper inappropriate for publication as an original research article in Nature Communications. The authors do not present any concrete theoretical results nor do they present any original experimental data. A review or commentary style article would appear more appropriate.

Reviewer #3:

Remarks to the Author:

The reply by Syzranov and Ramirez is strongly argued. I want to say from the outset that I do find this study interesting and thought provoking, and that alone probably means that it is worth publishing.

I want to bring to the attention of the authors one nagging counter example. The magnetically intercalated TMDs (materials like $M_x \text{NbS}_2$, $M = \text{Cr, Co, Mn, Fe, Ni}$ etc.) have an ordered magnetic lattice at $x=1/3$. Defects can take the form of vacancies or interstitials about this composition, and have been studied since the 70's (see Parkin and Friend *Physica* 99B (1980) 219-223). More recently, their spin-glass properties have been studied in the context of spintronics for both kinds of defects (DOI: 10.1126/sciadv.abd8452, *Nature Physics* volume 17, pages525–530 (2021)). These systems are not very frustrated (geometrically) and not very 2D so they should be in the conventional category. (They have strong interplanar exchange and even though they do appear to have two competing order parameters they are definitely nowhere near a spin liquid <https://doi.org/10.48550/arXiv.2106.01341>)

For dilute concentrations, these materials show the same trends as the GF systems discussed in this paper (increasing $\chi(T_g)$ and decreasing T_g). However, for excess concentrations that have interstitial defects, the trend is the same as the conventional systems in this paper (decreasing $\chi(T_g)$ for decreasing T_g). Perhaps I am mistaken, but I thought the natural explanation was the changing average distance between the magnetic ions would change the *average* exchange in opposite directions for the two cases. In dilute concentrations the average exchange would become closer to zero, but in excess compositions exchange would grow (or become more negative). Even in a simple Bleaney-Bowers model of susceptibility (B. Bleaney, K. D. Bowers, *Proc. R. Soc. Lond.* 1952, A214, 451), this would lead to the trend observed by the authors, at least for small changes in T_g .

I do not wish for the authors to explain every magnetic spin glass in existence! I think this trend is interesting, but I think the authors should be more open that there may be a simple explanation without the need to invoke a "hidden energy scale." Rather this could all be due to the complex statistical mechanics of disordered systems. This would lead to a trend that is different in systems that are non-magnetic but contain magnetic impurities than systems that are intrinsically magnetic lattices and contain vacancies. The case of magnetic lattices with excess magnetic ions may in this case be quite instructive.

So overall, I remain very skeptical of the "hidden" energy scale, but I do see that this observation is intriguing and therefore interesting to the community. A few more words of caution in the manuscript and I would be happier about accepting it for *Nat Comm*.

Replies to Reviewer #1

The authors have given detailed responses to the various points raised in the previous round of review. Many points are matters of differences of opinion that are unlikely to be resolved during further rounds of review. My sense from my own reading, the responses provided, and from the reviews from and responses to the other referees is that many of the ideas presented here will be considered interesting yet contentious within the community. I am happy to suggest that the paper be published.

We are grateful to the Reviewer for refereeing our manuscript and for the positive assessment of our work.

Replies to Reviewer #2:

In this manuscript, Syznarov and Ramirez, compare the properties of geometrically frustrated spin glasses with conventional spin glasses using a collection of susceptibility data assembled from the literature. I have reviewed a previous version of this manuscript when it was under consideration at Nature Physics and have now read the revised version as well as the authors responses. The authors show that in geometrically frustrated spin glasses, a general trend can be observed where as the magnetic sublattice is diluted the freezing transition is suppressed in temperature, while the susceptibility magnitude increases.

In my previous report, I commented on the use of the term defects and disorder to describe what I feel is most accurately labelled as magnetic dilution. The manuscript still paints the overwhelming picture that the leading effect of replacing a magnetic ion with a non-magnetic ion is disorder. See for example the paragraph that starts at line 48. The effect of magnetic dilution is completely disregarded and never addressed and is extremely important to interpreting the trends discussed here. Further, the use of the term defects is still misleading, particularly in the context of Fig 1, where it appears in panel (a) and (b) with opposite meaning. Putting aside semantic differences, my primary criticism does not relate to the choice of terms but rather the fact that the effect of dilution (or whatever term the authors prefer) – meaning the reduced connectivity of the magnetic lattice – is never addressed.

In our view, the criticism of the referee indeed concerns the semantics used in the paper. A significant portion of the paper describes observations following from experimental data, before discussing the mechanisms for the observed trends. It does not *disregard* dilution or any other effect, contrary to the referee's claim.

Nevertheless, to improve the clarity of the paper to the largest possible audience and eliminate possible misinterpretation, we changed, following the referee's comments, a number of terms relating to the concepts of disorder and defects:

- In Figs. 1 (a) and (b), the labels “defect density” are replaced by “magnetic impurity density” and “vacancy density”, respectively. This emphasises different nature of the defects in non-GF and GF systems and leaves no room for confusion.
- We rewrote the last but one paragraph in the introduction, substituting the word “disorder” by “defects” and emphasising the said different nature of defects in GF and non-GF systems.
- Minor modifications throughout the manuscript emphasising that quenched disorder in GF systems comes, at least in part, from vacancy defects.

The other main finding of this study is that, in the systems surveyed, the magnitude of the anomaly at T_g is increasing as a function of dilution/disorder. I agree that this is an interesting observation but again there are many conflating factors. One key factor is that dilution by reducing connectivity, alleviates geometric frustration. In the least diluted samples (the lowest defects in the authors terms), the behavior is closest to the ideal spin liquid limit where a larger fraction of the spins remain dynamic (as has been seen in other probes such as the persistent spin dynamics seen in μ SR). Then, as magnetic ions are replaced, frustrated is alleviated and the magnet becomes more conventional, resulting in more spins participating in the freezing transition and therefore enhancing the strength of the susceptibility anomaly. In the final section of the paper, titled “Online methods” the authors present a picture for the behavior described above. The scenario being presented is vague and insufficiently detailed.

In the manuscript, the qualitative picture of reducing the glass transition temperature due to diluting the bulk spins has already been presented in the paragraph “Effective degrees of freedom” in the “Results” section (not in “Online methods”). It was also noted that this picture cannot explain the dependence of the magnetic susceptibility on the vacancy concentration.

Nevertheless, to further improve the clarity of the revised manuscript and following the referee’s comments, we rewrote this paragraph to emphasise the effect of dilution that the referee describes, together with the need for further theories and experiment to reconcile this effect with the observed behaviour of the magnetic susceptibility.

The existence of frustrated materials without a spin glass transition is handwaved and relegated to the supplemental material.

In the manuscript, we make a statement (at the end of the “Methods” section) about the absence of finite-temperature spin-glass transitions in 2D. It is not “handwaved”. Such an absence has been rather reliably established in multiple numerical works. At the same time, there is no analytical argument for such an absence. Therefore, when making a statement about such an absence in the main text, we cite, as an example, the most recent and accurate numerical study [46] verifying the statement. In Supplemental Material, we clarify it and cite multiple additional papers on the topic. Because the statement has no direct bearing on the results of the manuscript, Supplemental Material is an appropriate place for such a clarification. In the main text we added a reference to Supplemental Material at the respective phrase.

The reply by Syzranov and Ramirez is strongly argued. I want to say from the outset that I do find this study interesting and thought provoking, and that alone probably means that it is worth publishing.

We thank the referee for the careful reading of the manuscript, positive review and helpful comments. Below, we respond to these comments.

I want to bring to the attention of the authors one nagging counter example. The magnetically intercalated TMDs (materials like $M_x\text{NbS}_2$, $M = \text{Cr, Co, Mn, Fe, Ni}$ etc.) have an ordered magnetic lattice at $x=1/3$. Defects can take the form of vacancies or interstitials about this composition, and have been studied since the 70's (see Parkin and Friend *Physica* 99B (1980) 219-223). More recently, their spin-glass properties have been studied in the context of spintronics for both kinds of defects (DOI: 10.1126/sciadv.abd8452, *Nature Physics* volume 17, pages525–530 (2021)). These systems are not very frustrated (geometrically) and not very 2D so they should be in the conventional category.(They have strong interplanar exchange and even though they do appear to have two competing order parameters they are definitely nowhere near a spin liquid <https://doi.org/10.48550/arXiv.2106.01341>)

For dilute concentrations, these materials show the same trends as the GF systems discussed in this paper (increasing $\chi(T_g)$ and decreasing T_g). However, for excess concentrations that have interstitial defects, the trend is the same as the conventional systems in this paper (decreasing $\chi(T_g)$ for decreasing T_g). Perhaps I am mistaken, but I thought the natural explanation was the changing average distance between the magnetic ions would change the *average* exchange in opposite directions for the two cases. In dilute concentrations the average exchange would become closer to zero, but in excess compositions exchange would grow (or become more negative). Even in a simple Bleaney-Bowers model of susceptibility (B. Bleaney, K. D. Bowers, *Proc. R. Soc. Lond.* 1952, A214, 451), this would lead to the trend observed by the authors, at least for small changes in T_g .

With regard to the mentioned “counter example”, however, we would like to respectfully disagree with the referee. As they point out, the Fe-intercalated transition metal dichalcogenide (TMD) $\text{Fe}_{0.33}\text{NbS}_2$ is an antiferromagnet with $T_N = 45\text{K}$. For lower Fe concentrations, this system exhibits spin glass-like freezing. At right is reproduced a figure from the arXiv paper cited by the referee, with our annotations in red. One can see that, contrary to the referee’s statement, the glass temperature (T_f) increases as Fe concentration (x) is reduced from 0.32 to 0.31. Moreover, we do not see clear evidence that $\chi(T_g)$ increases as T_f decreases, which is what we show for strongly geometrically frustrated systems. Indeed, for $x = 0.32$, it appears as though the peak in $\chi(T)$ is a vestige of the AF order, likely due to finite-size regions of well-ordered Fe atoms.

FIG. 11. Magnetization measurements for $x =$ (a) 0.33, (b) 0.32 and (c) 0.31 with applied field along c and in ab -plane. The dashed and solid lines corresponding to the measurements with field-cooled and zero field cooled process. The measurements are used from the same sample in the neutron diffraction experiment.

A second set of data, shown below, are found in the Supplementary information of the Nature Physics article cited by the referee. In these data we see two compounds with Fe concentrations below $x = 0.33$ and peaks that resemble spin glass cusps. It is true that $\chi(T_g)$ increases as x decreases, but clearly T_g does not decrease as x decreases from 0.31 to 0.32.

This spin glass behaviour in Fe_xNbS_2 and related compounds may be interesting to explore further, guided by the theoretical framework presented in our manuscript, especially since the Fe lattice is

triangular and therefore might reveal the physics of the eminent phase. At present, however, we do not think the data for this system are complete enough to argue one way or another.

I do not wish for the authors to explain every magnetic spin glass in existence! I think this trend is interesting, but I think the authors should be more open that there may be a simple explanation without the need to invoke a "hidden energy scale." Rather this could all be due to the complex statistical mechanics of disordered systems. This would lead to a trend that is different in systems that are non-magnetic but contain magnetic impurities than systems that are intrinsically magnetic lattices and contain vacancies. The case of magnetic lattices with excess magnetic ions may in this case be quite instructive.

So overall, I remain very skeptical of the "hidden" energy scale, but I do see that this observation is intriguing and therefore interesting to the community. A few more words of caution in the manuscript and I would be happier about accepting it for Nat Comm.

We agree with the Reviewer's sentiment that due caution should be exercised when discussing new and unexpected observations following from experimental data. In principle, the term "hidden energy scale" in the manuscript is a label for an observed energy scale and, on its own, does not imply any incredible scenario for its existence.

Nevertheless, to exercise due caution, we briefly summarised at the beginning of the "Methods" section possible scenarios for the hidden energy scale that we envision, while emphasising that identifying a particular microscopic mechanism requires further theoretical and experimental studies. In a big portion of that section, we focus on the scenario that we believe to be the most likely, but we agree that the readers can judge for themselves the likelihood of each scenario.